# Cell-free chromatin particles released from dying host cells are global instigators of endotoxin sepsis in mice

**Indraneel Mittra** *, **Kavita Pal, Namrata Pancholi, Pritishkumar Tidke, Sophiya Siddiqui, Bhagyeshri Rane, Jenevieve D'souza, Alfina Shaikh, Saili Parab, Sushma Shinde, Vishal Jadhav, Soniya Shende, Gorantla V. Raghuram**

Translational Research Laboratory, Advanced Centre for Treatment, Research and Education in Cancer, Tata Memorial Centre , Kharghar, Navi Mumbai, India

* imittra@actrec.gov.in

**Data Availability Statement:** All relevant data are within the paper and its Supporting Information files.

## Abstract

We have earlier reported that cell-free chromatin (cfCh) particles that are released from dying cells, or those that circulate blood, can readily enter into healthy cells, illegitimately integrate into their genomes and induce dsDNA breaks, apoptosis and intense activation of inflammatory cytokines. We hypothesized that sepsis is caused by cfCh released from dying host cells following microbial infection leading to bystander host cell apoptosis and inflammation which are perpetuated in a vicious cycle with release of more cfCh from dying host cells. To test this hypothesis we used three cfCh inactivating agents namely 1) anti-histone antibody complexed nanoparticles which inactivate cfCh by binding to histones; 2) DNase I which inactivates cfCh by degrading its DNA component, and 3) a novel pro-oxidant combination of Resveratrol and Copper which, like DNase I, inactivates cfCh by degrading its DNA component. Female C57 BL/6 mice, 6–8 weeks old, were administered a single i.p. injection of LPS at a dose of 10 mg/Kg or 20 mg/Kg with or without concurrent treatment with the above cfCh inactivating agents. Administration of cfCh inactivating agents concurrently with LPS resulted in prevention of following pathological parameters: 1) release of cfCh in extra-cellular spaces of brain, lung and heart and in circulation; 2) release of inflammatory cytokines in circulation; 3) activation of DNA damage, apoptosis and inflammation in cells of thymus, spleen and in PBMCs; 4) DNA damage, apoptosis and inflammation in cells of lung, liver, heart, brain, kidney and small intestine; 5) liver and kidney dysfunction and elevation of serum lactate; 6) coagulopathy, fibrinolysis and thrombocytopenia; 7) lethality. We conclude that cfCh that are released from dying host cells in response to bacterial endotoxin represents a global instigator of sepsis. cfCh inactivation may provide a novel approach to management of sepsis in humans.

## Introduction

Sepsis is a common and lethal syndrome with a hospital death rate of 40–50% [1]. It has been estimated that 30 million people are affected by sepsis and 6 million succumb to it globally

**Funding:** This study was supported by the Department of Atomic Energy, Government of India, through its grant CTCTMC to Tata Memorial Centre awarded to I.M. The funders had no role in study design, data collection and analysis, decision to publish, or preparation of the manuscript.

**Competing interests:** The authors have declared that no competing interests exist.

every year [2]. Sepsis is a complex disorder characterized by: a hyper-inflammatory state with marked release of inflammatory cytokines in circulation [3]; immune paralysis due to apoptosis of lymphoid cells, especially of thymus and spleen [4]; inflammation and apoptosis of parenchymal cells leading to multi-organ dysfunction, especially of kidney and liver [5]; disseminated intravascular coagulation [6] and deposition of fibrin causing micro-vascular thrombi which exacerbate organ dysfunction [7]; consumption of clotting factors leading to fibrinolysis and life-threatening haemorrhage [8].

In spite of intensive research, patho-physiology of sepsis remains poorly understood hindering development of effective therapies [9]. Numerous trials of anti-inflammatory drugs having failed [10], a global clarion call has been sounded for new approaches to treatment of sepsis [11]. Herein, we provide one such new approach that involves inactivating cell-free chromatin (cfCh) particles that are released from dying host cells following severe microbial infection. We have earlier reported that cfCh particles that are released from dying cells, or those that circulate blood, can readily enter into healthy cells, illegitimately integrate into their genomes and induce dsDNA breaks, apoptosis and intense activation of inflammatory cytokines [12–17]. The uptake of cfCh particles by cells was found to be rapid and spontaneous. *In vitro* experiments in which mouse fibroblast cells were co-cultured with dying cells, maximum uptake of cfCh released from the dying cells was reached at 6 h, and microarray analysis at this time point showed up-regulation of pathways related to phagocytosis, suggesting a possible mechanism by which cfCh are ingested by cells [12]. The intracellular cfCh associated themselves with host cell chromosomes followed by their genomic integration [13, 12]. The latter involved dsDNA breaks as indicated by activation of H2AX and repair of the integrated cfCh particles by non-homologous end joining [13, 12, 14]. The extensive DNA damage also evoked activation of apoptotic pathways leading to death of a proportion of cells [13, 12, 14]. Surprisingly, genomic integration of cfCh and the resulting dsDNA breaks triggered marked activation of inflammatory cytokines to include NFκB, IL-6, IFNγ and TNFα [13, 15, 16]. Fluorescent NFκB signals were found to co-localise with those of γH2AX suggesting that inflammation is a direct response to dsDNA breaks [13, 15, 16]. In summary, cfCh from dying cells, or those that circulate in blood, can lead to extensive DNA damage, apoptosis and inflammation in healthy cells [17].

Based on the above findings, we hypothesized that sepsis may be caused by release of cfCh from dying host cells that follow microbial infection to trigger DNA damage, apoptotic and inflammatory responses in healthy cells of the host. The released cfCh from dying cells trigger a vicious cycle with release of more cfCh from dying host cells thereby perpetuating and amplifying the pathological complications of sepsis. cfCh released from dying host cells as a cause of sepsis would be consistent with the consensus definition of the International Sepsis Forum as "a life-threatening condition that arises when the body's response to an infection injures its own tissues and organs" [18]. Herein we show, in the mouse endotoxin model, that three different agents that have the ability to inactivate cell-free chromatin (cfCh), namely: 1) anti-histone antibody complexed nanoparticles (CNPs) which inactivates cfCh by binding to histones; 2) DNase I which inactivates cfCh by degrading its DNA components and 3) a novel pro-oxidant combination of Resveratrol and Copper (R-Cu) which, like DNase I, can degrade the DNA components of cfCh prevent multiple pathological parameters of sepsis following intra-peritoneal administration of LPS leading to improved survival of mice.

## Materials and methods

### Aim, design and setting of the study

The aim of the study is to investigate whether cfCh inactivating agents would prevent LPS induced sepsis. In a pre-clinical setting, mice were given i.p. injections of LPS with and without

concurrent administrations of cfCh inactivating agents such as 1) anti-histone antibody complexed nanoparticles (CNPs); 2) DNase I, and 3) a combination of Resveratrol and Copper (R-Cu), and various pathological parameters of sepsis were estimated at appropriate time points.

## Animal ethics approval

The experimental protocol was approved by the Animal Ethics Committee of Advanced Centre for Treatment, Research and Education in Cancer, Tata Memorial Centre, Navi Mumbai, India under two projects: the 1st project was entitled 'To evaluate the ability of 1) Resveratrol-$Cu^{2+}$, 2) anti-histone antibody complexed nanoparticles (CNPS) and 3) DNase I preventing lethality induced by lipopolysaccharide (LPS)' (Project no. 20/2016); the 2nd project was entitled 'To evaluate the ability of 1) Resveratrol-$Cu^{2+}$, 2) anti-histone antibody complexed nanoparticles (CNPS) and 3) DNase I in preventing tissue toxicity induced by Lipopolysaccharide (LPS)' (Project no. 2/2017). The experiments were carried out according to the Committee's animal safety guidelines and ARRIVE guidelines.

ACTREC IAEC maintains that the respectful treatment, care and use of animals involved in research is an ethical and scientific necessity and that the use of animals in research and teaching contributes to the advancement of knowledge and the acquisition of understanding. All medical and biological scientists involved in this study have undergone training in ethical treatment and management of animals under supervision of attending veterinarian. They affirm that respect for all forms of life is an inherent characteristic of biological and medical scientists who conduct research involving animals.

## Animals

We used inbred female C57Bl/6 mice obtained from the Institutional Animal Facility for our study. All mice were maintained in agreement with Institutional Animal Ethics Committee (IAEC) standards. Mice of age 6–8 weeks were randomly assigned to control and experimental groups. All animals had free access to water and food. They were housed in pathogen-free cages containing husk bedding under 12-h light/dark cycle. HVAC system was used to provide a controlled room temperature, humidity and air pressure. The study involved three experiments 1) Effect of cfCh inactivating agents in reducing the surge in chromatin levels and inflammatory cytokines in serum after 18h of LPS administration, 2) Effect of cfCh inactivating agents in reducing tissue DNA damage, apoptosis and inflammation after 72h administration of LPS and 3) Effect of cfCh inactivating agents in reducing the mortality induced by LPS. All experiments were carried out at a sub lethal LPS dose of 10 mg/Kg except for the survival study in which a dose of 20 mg/Kg was used. The various experiments were either of 18 h or of 72 h duration, with one experiment on lethality lasting for 10 days. Humane endpoints were defined as reduced physical activity and weight loss. These parameters were not assessed in experiments lasting 18 hours; however, no visible loss of activity was observed. For experiments lasting 72 h, both weight and activity of animals were recorded (S1 Table). The experiment that lasted for 10 days to evaluate the effects of cfCh inactivating agents in preventing lethality following LPS treatment, detailed record of weight and activity were kept and are given in S2A and S2B Table. At the end of each study, the animals were anaesthetized under isoflurane in a fume hood and blood was collected. In certain studies, organs were collected after CO2 euthanization followed by cervical dislocation under the supervision of FELASA certified attending veterinarian. Any animal reaching humane endpoints were euthanized using CO2 inhalation followed by cervical dislocation under the supervision of attending veterinarian, irrespective of the control or test groups.

## Ethical approval and consent to participate

Our study does not involve human subjects; hence ethical approval and patient consent is not applicable to our manuscript.

## Materials

Details of analytical kits and antibodies used in this study are given in S3 Table. LPS of Salmonella enteric serotype *typhimurium* was obtained from Sigma, USA (catalogue no. L6511). LPS was dissolved in PBS and administered as a single i.p. injection.

## Preparation of cfCh inactivating agents, their sources, dosage and routes of administration

We used three cfCh inactivating agents in our study, namely: 1) anti-histone antibody complexed nanoparticles (CNPs); 2) DNase I, and 3) a combination of Resveratrol and Copper (R-Cu). 1) CNPs were prepared according to our earlier report [19,20] except that histone H4 IgG was exclusively used to prepare CNPs (S3 Table). CNPs, 50 μg in 100 μl of buffer, was administered once a day i.p; the 1st CNPs dose being administered four hours prior to LPS administration. 2) DNase I (Sigma-Aldrich; Catalogue No- DN25-1G) dissolved in saline was administered at a twice daily dose of 15 mg/kg i.p. The 1st dose of DNase I was administered 4 h prior to LPS treatment. 3) The plant polyphenol Resveratrol (R) is an antioxidant which has been extensively researched for its health promoting properties [21]. R can paradoxically act as a pro-oxidant in presence of copper (Cu) because its ability to reduce Cu (II) to Cu (I) resulting in the generation of free radicals which can cleave plasmid DNA [22, 23]. We have shown that R-Cu can not only cleave, but also degrade, genomic DNA *in vitro*.

Light metals, such as Zn, are not active when combined with resveratrol (unpublished data). The DNA degrading activity of R-Cu is maintained even when the molar concentration of Cu is reduced more than 10,000 fold with respect to that of R [24]. We have also shown that R-Cu is active *in vivo*, and can degrade the DNA component of cfCh thereby inactivating it even when the molar ratio of R:Cu is kept at 1: ¯10 000 [20, 25]. The final concentration of R in the current study was 1 mg/kg and that of Cu was 0.1 μg/kg, i.e. at a final ratio of $1:10^{-4}$. R and Cu were administered orally separately one after the other. The 1st dose of R-Cu was administered four hours prior to LPS challenge. The sources of R and Cu were (Resveratrol, Trade name—TransMaxTR, Biotivia LLC, USA; Copper, Trade name—Chelated Copper, J.R. Carlson Laboratories Inc. USA).

## Detection of cfCh generated in vital organs by fluorescence Immune-staining and confocal microscopy

Animals were injected with LPS with or without cfCh inactivating agents. Animals (2 in each group) were sacrificed after 72 h and brain, lung and heart were harvested and fixed in formalin. Unstained FFPE slides were processed for dual immune-staining using antibodies against DNA (1:100 dilution) (Novus Biologicals LLC, Littleton, USA, Catalogue No. NB110-89473) and histone H4 IgG (1: 100 dilution) (custom synthesised by Bioklone Biotech Pvt Ltd, Chennai, India). The secondary antibody used was Rhodamine labelled anti-mouse antibody (red, in case of DNA) (Merck Millipore, USA, Catalogue No. AP160R) and FITC labelled anti-rabbit antibody (green, in case of histone H4) (Abcam®, UK, Catalogue No. ab6717). It should be pointed out that while the primary antibody against DNA was mouse IgM (Novus Biologicals LLC, Littleton, USA, Catalogue No. NB110-89473), the secondary antibody was IgG, but which also reacts with the light chains from all Mouse immunoglobulin classes (including

mouse IgM) (Sigma-Aldrich, AP160R). Slides were mounted with vectashield DAPI mounting medium (Vector Laboratories, Catalog#H-1200). Immune-stained slides were examined by confocal microscopy and images were captured in LSCM780 (Carl-Zeiss AG, Oberkochen, Germany) microscope and analysed in ZEN 2012 software version 8.1. cfCh in vital organs of mice were identifiable by co-localizing fluorescence signals of DNA (red) and histone H4 (green) generating yellow / white coloured particles. Fluorescence intensity of six randomly chosen confocal fields (~ 150 cells) from each slide was recorded and the mean (± S.E.M) fluorescence intensity (MFI) was calculated.

### Estimation of cfCh in serum

This estimation were performed at 18 h post LPS. Blood was obtained by orbital puncture followed by separation of serum by standing at room temperature for 3 h without centrifugation. cfCh estimation was performed as described by us earlier using the Cell Death Detection ELISAPLUS kit [20, 25]. Results were expressed in arbitrary units in the form of absorbance value in spectrophotometer.

### Estimation of serum inflammatory cytokines in serum

These estimations were performed at 18 h post LPS. Blood was obtained by orbital puncture followed by separation of serum by standing at room temperature for 3 h. followed by centrifugation at 1800 x g for 10 min. Estimation of CRP, IL-6, IL-1β, TNF-α and IFN-γ were performed by ELISA according to vendors' instructions as described by us earlier [20].

### WBC and platelets count

These estimations were performed at 18 h post LPS. Total WBC and Platelet counts were estimated in the blood collected in EDTA chelated tubes and processed immediately for the cell count using the ADVIA® 2120i (Siemens) automated machine.

### Detection of cellular DNA-damage, apoptosis and inflammation in organs, tissues and PBMCs by immuno-fluorescence

These estimations were performed at 72 h post LPS. Organs were snap frozen and immune-staining was done on cryosections. For PBMC, blood was collected by orbital puncture in EDTA chelated tubes. PBMCs were separated using Ficoll gradient centrifugation technique and smears were prepared on slides. Analysis of γH2AX, active caspase-3, Bcl-2, NF-κB, IL-6, TNF-α and IFN-γ was done by indirect immuno-fluorescence as described by us earlier [20]. In brief, PBMC or cryo-sectioned slides were washed with PBS and fixed in 4% paraformaldehyde for 20 minutes followed by permeabilization with 0.2% Triton X-100 at room temperature for 30 minutes. Slides were washed thrice with 1X PBS for 5 minutes each and blocked using Bovine Serum Albumin (3% (BSA) at room temperature for 1 h. The sections were immuno-stained with primary antibody specific for γH2AX, active caspase-3, Bcl-2, NF-κB, IL-6, TNF-α and IFN-γ (in 3% BSA) at 4˚C overnight followed by secondary antibody (in 3% BSA) incubation in dark at room temperature for 1 h. After three successive 1x PBS washes, tissue sections were stained with DAPI containing mounting medium (VECTASHIELD, Vector laboratories). Images were acquired and analysed under Applied Spectral Bio-imaging system (Applied Spectral Imaging, Israel). Mean fluorescence intensity (MFI) was determined from 1000 cells for each animal and final results were expressed as mean (± S.E.M.).

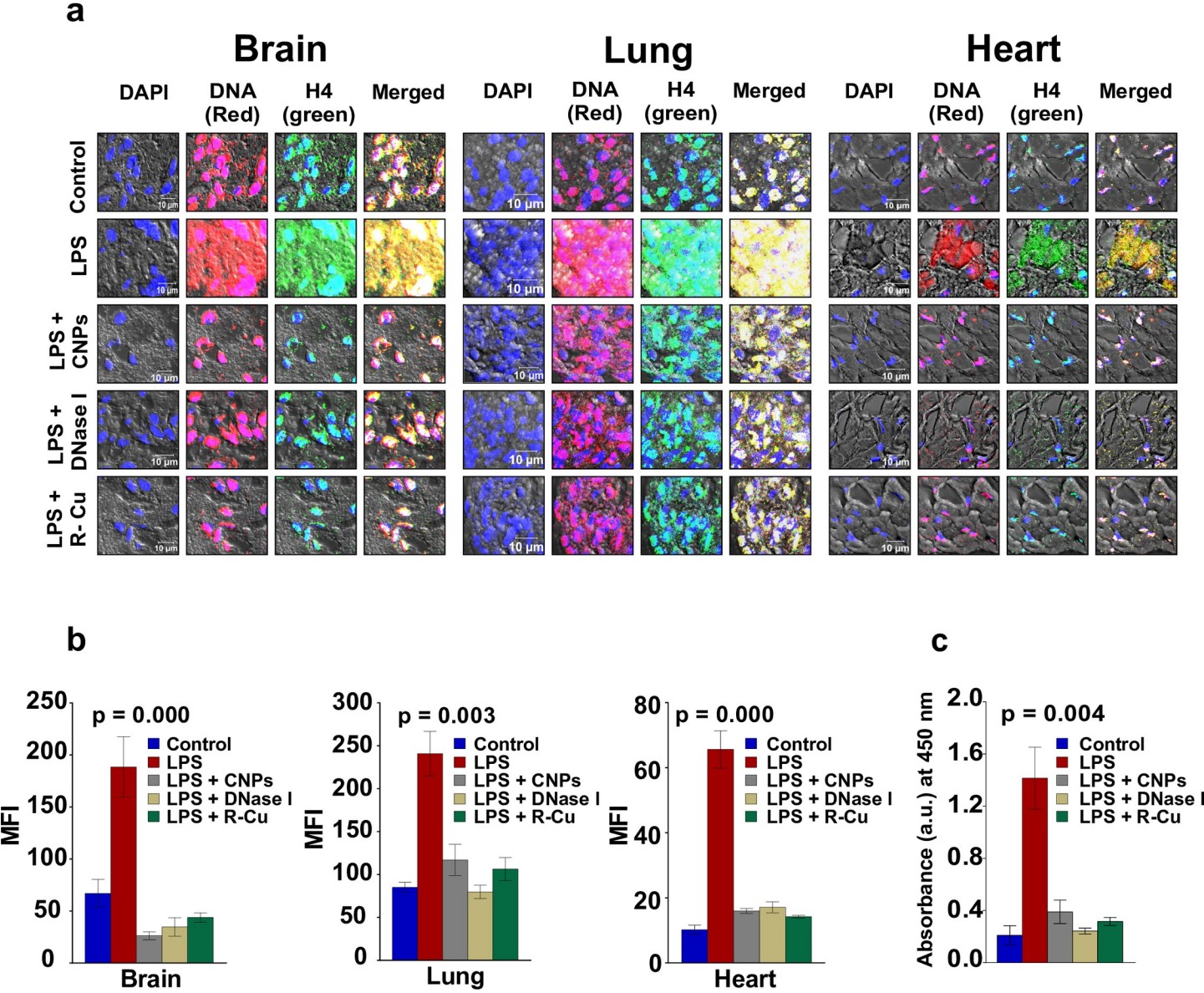

**Fig 1. Release of cfCh into extra- nuclear spaces of vital organs and into the circulation following LPS treatment and its prevention by cfCh inactivating agents. a.** Fluorescence immuno-staining and confocal microscopy images of sections of mouse brain, lung and heart stained with fluorescent antibodies against DNA and histone H4 and examined by confocal microscopy. Co-localizing DNA (red) and histone H4 (green) fluorescent signals generate yellow / white coloured particles representing cfCh. Many yellow particles are seen outside the nucleus in the intra- / extracellular spaces in control animals with dramatic increase following LPS treatment. Treatment with CNPs, DNase 1 and R-Cu markedly reduced the number of yellow particles. **b.** Graphical representation of cfCh release into extracellular spaces of vital organs and its prevention by CNPs, DNase 1 and R-Cu. Six confocal fields were randomly captured (~150 cells) and their fluorescence intensities were recorded. Each group comprised of 2 animals and the histograms provide mean (± SEM) MFI values in each case. **c.** Release of cfCh into the circulation following LPS treatment and its prevention by cfCh inactivating agents. Serum cfCh was estimated using Cell Death Detection ELISA. Results (mean ±SE) are expressed in arbitrary units (a.u.) of absorbance values detected by spectrophotometry. Each group comprised of 5 animals and the histogram depicts mean (± SEM) values. Mean (± SEM) values of N = 5 between groups were compared using non parametric one-way ANOVA (Kurskal—Wallis test) with Dunn's multiple comparison method at the significance and confidence level of p = 0.05.

## Serum biochemical analysis for measurement of organ dysfunction in serum

These estimations were performed at 72 h post LPS. Blood was obtained by orbital puncture under anaesthesia, followed by separation of serum by standing at room temperature for 3 h

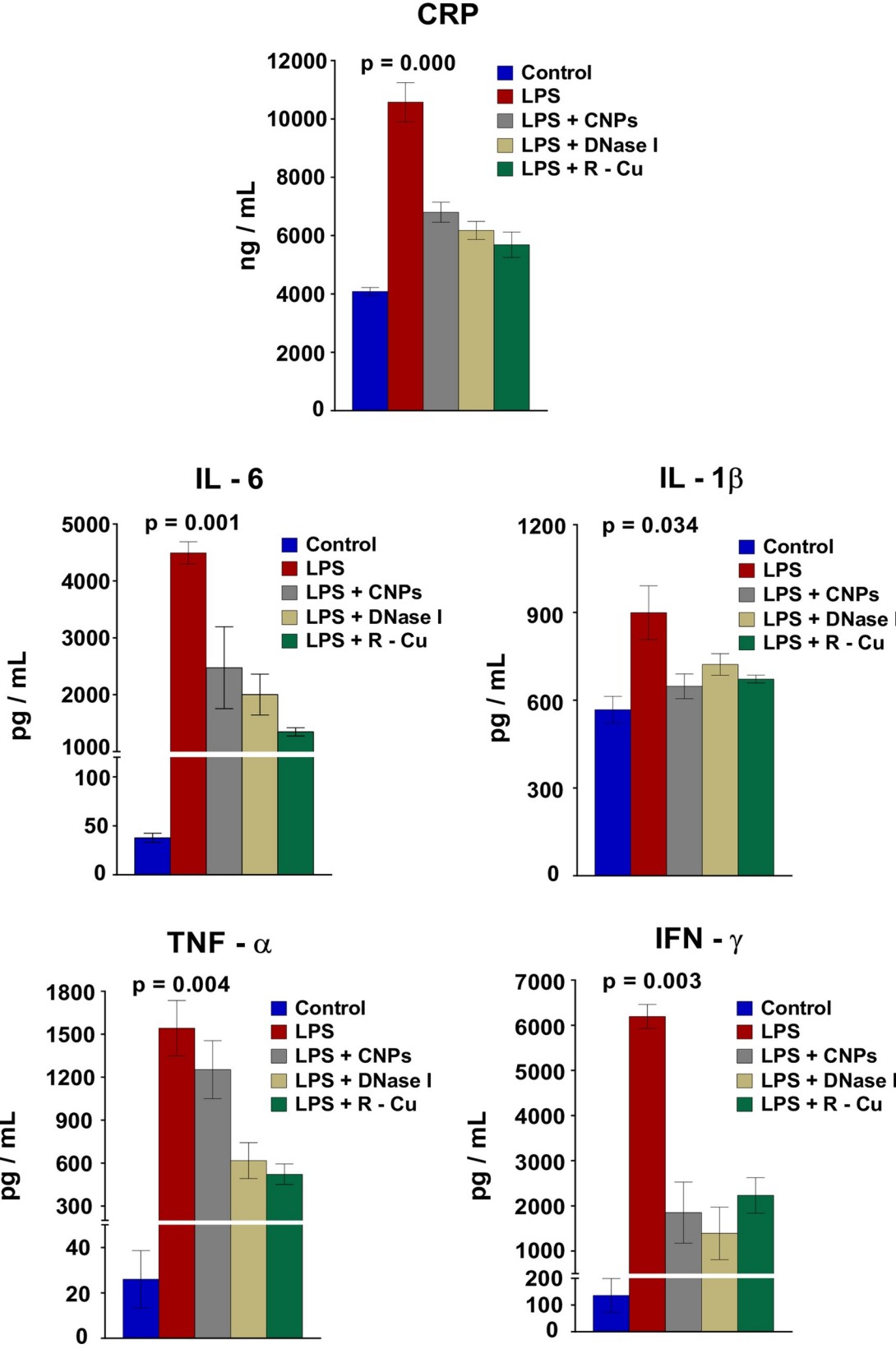

**Fig 2. Release of inflammatory cytokines into the circulation following LPS treatment and its prevention by cfCh inactivating agents.** LPS treatment resulted in marked increase in release of various inflammatory cytokines which were significantly reduced by concurrent treatment with CNPs, DNase 1 and R-Cu. Cytokines were estimated by ELISA at 18 h post LPS. Methodological details are given under Material and Methods section. Each group in all experiments comprised of 5 animals and the histograms provide mean (± SEM) values. Mean (± SEM) values of N = 5 between groups were compared using non parametric one-way ANOVA (Kurskal—Wallis test) with Dunn's multiple comparison method at the significance and confidence level of p = 0.05.

followed by centrifugation at 1800 x g for 10 min. Alanine aminotransaminase (ALT), aspartate transaminase (AST), lactate dehydrogenase (LDH), creatinine and blood urea nitrogen (BUN) levels were estimated using the automated Dimension EXL with LM machine (Siemens). Serum lactate was estimated by colorimetric assay kit (S3 Table).

## Blood plasma analysis for coagulopathy

These estimations were performed at 72 h post LPS. Blood was collected by orbital puncture in EDTA tubes followed by centrifugation at 1800 x g for 10 min. Fibrinogen, antithrombin, protein C and TAT complex were estimated in plasma using ELISA kits (S3 Table) and per vendors' instructions. Fibrinogen deposition in cryo-sectioned liver sections was performed by indirect Immuno-fluorescence using anti-fibrinogen antibody at 1:200 dilution. Goat anti-mouse FITC conjugated secondary antibody was used at concentration of 1:500. Slides were mounted with vectashield DAPI mounting medium. The immuno stained slides were examined by fluorescence microscope under 40X magnification and images were captured. Thousand cells per slide were counted for percent positive cells and mean ± SEM was calculated.

## Survival analysis

Forty animals were divided into 4 groups of 10 mice each and all groups received a lethal dose of LPS (20 mg/kg). CNPs, DNase I and R-Cu were administered in doses and frequencies as described above. The cfCh inactivating agents were commenced 4 h prior to LPS injection. Survival between groups was compared by Kaplan–Meier survival analysis using log-rank test.

## Statistical analysis

Statistical analyses were done using GraphPad Prism 6 (GraphPad Software, Inc., USA. Version 6.0). Mean (± SEM) values of N = 5 between groups were compared using non parametric one-way ANOVA (Kurskal—Wallis test) with Dunn's multiple comparison method at the significance and confidence level of p = 0.05. Comparisons of control and treatment groups were made with LPS treated groups separately for each organ/tissue. Kaplan-Meier curves were used to express survival rates and survival curves were compared using log-rank test with the use of PRISM Ver.6.0.

# Results

## Release of cfCh into extracellular spaces of vital organs and in circulations and its prevention

LPS administration is known to cause tissue injury and cellular apoptosis via the production of free radicals and inflammatory cytokines [26, 27]. We investigated by fluorescence immune-staining and confocal microscopy using antibodies against DNA and histone H4, if cellular apoptosis following LPS challenge would lead to release of cfCh into extracellular spaces of vital organs, namely brain, lung and heart. cfCh could be identified as yellow / white particles resulting from co-localizing fluorescent signals of DNA (red) and histone H4 (green) (Fig 1A).

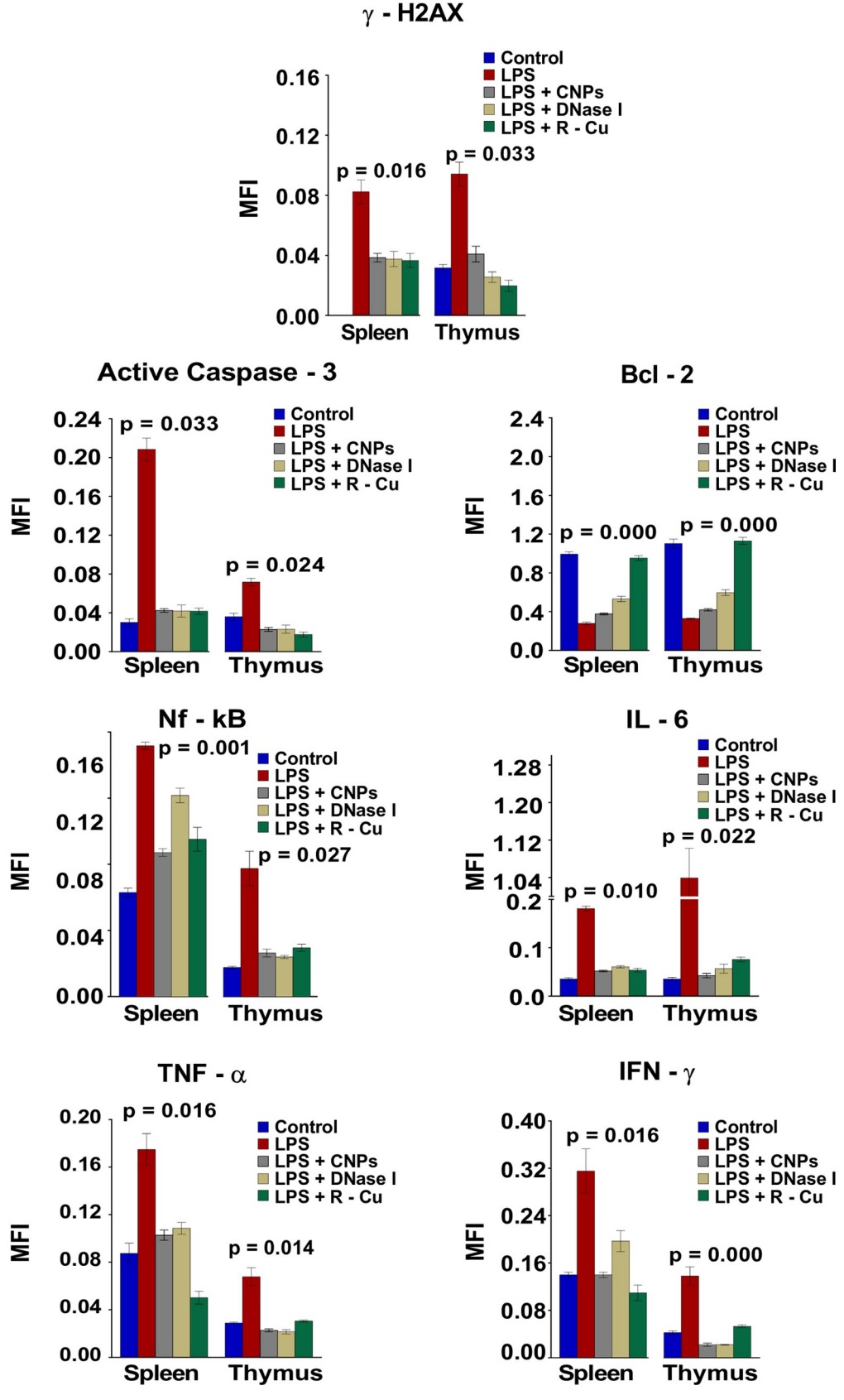

**Fig 3. DNA damage, apoptosis and inflammation in in cells of spleen and thymus following LPS treatment and their prevention by cfCh inactivating agents.** Treatment with LPS resulted in marked increase in DNA damage, apoptosis and inflammation in cells of spleen and thymus which were significantly reduced by concurrent treatment with CNPs, DNase 1 and R-Cu. The above parameters were estimated by indirect immuno-fluorescence performed at 72 h post LPS. Methodological details are given under Material and Methods section. Each group in all experiments comprised of 5 animals and the histograms provide mean (± SEM) values. Mean (± SEM) values of N = 5 between groups were compared using non parametric one-way ANOVA (Kurskal—Wallis test) with Dunn's multiple comparison method at the significance and confidence level of p = 0.05.

It should be noted that in some DAPI positive areas of the nuclei are not stained with DNA and / or histone H4 antibodies. This is likely to be due to unevenness of cut surfaces of paraffin sections resulting in the antibodies not being able to access these DNA / histone epitopes in the nuclei. Multiple dual labelled fluorescence particles were detectable in areas outside the nuclei of vital organs of untreated control mice which are strongly suggested of being cfCh particles (Fig 1A). It cannot be excluded, however, that some of these may represent non-specific binding of antibodies to dead tissues. However, following LPS treatment, copious effusion of cfCh particles into extra-nuclear areas was evident (Fig 1A). The efflux of cfCh could be virtually abolished by concurrent treatment with cfCh inactivating agents, viz anti-histone antibody complexed nanoparticles (CNPs), DNase I and R-Cu (Fig 1A and Fig 1B). LPS treatment also led to marked release of cfCh into the circulation which could also be prevented by the cfCh inactivating agents (Fig 1C).

## Release of inflammatory cytokines in circulation and its prevention

Inflammation is a hallmark of sepsis with release of inflammatory cytokines in circulation [27, 28]. A surge of CRP, IL-6, IL-1β, TNFα, IFNγ in blood occurred following LPS challenge which was significantly reduced by all three cfCh inactivating agents (Fig 2).

## DNA damage, apoptosis and inflammation in cells of thymus and spleen and their prevention

Apoptotic cell death of lymphocytes leading to immune paralysis is a major cause of death from sepsis [29] which is known to be associated with apoptosis of thymocytes and splenocytes [29]. A marked activation of H2AX following LPS treatment was seen in cells of thymus and spleen as evidence of DNA damage, and of apoptosis by up-regulation of active Caspase 3 with simultaneous down-regulation of BCL-2. Activation of all three parameters could be reversed by CNPs, DNase I and R-Cu (Fig 3 and S1A–S1G Fig).

LPS treatment also led to pronounced activation of inflammatory cytokines which are important triggers for lymphocyte apoptosis in thymus and spleen [30]. The up-regulated levels of NFκB, IL-6, TNF-α and INF-γ in cells of spleen and thymus were significantly reduced following concurrent treatment with cfCh inactivating agents (Fig 3). Peripheral blood mononuclear cells (PBMCs) also showed evidence of DNA damage, apoptosis and inflammation [31] following LPS administration which were abrogated by CNPs, DNase I and R-Cu (Fig 4, upper three panels and S2A and S2B Fig).

The depletion of total leukocyte count following LPS was also reversed by the three cfCh agents (Fig 4, lowest panel).

## DNA damage, apoptosis and Inflammation of parenchymal cells of vital organs and their prevention

Inflammation and apoptosis of parenchymal cells of vital organs leading to multi-organ dysfunction is another cardinal feature of sepsis [32]. We observed extensive DNA damage,

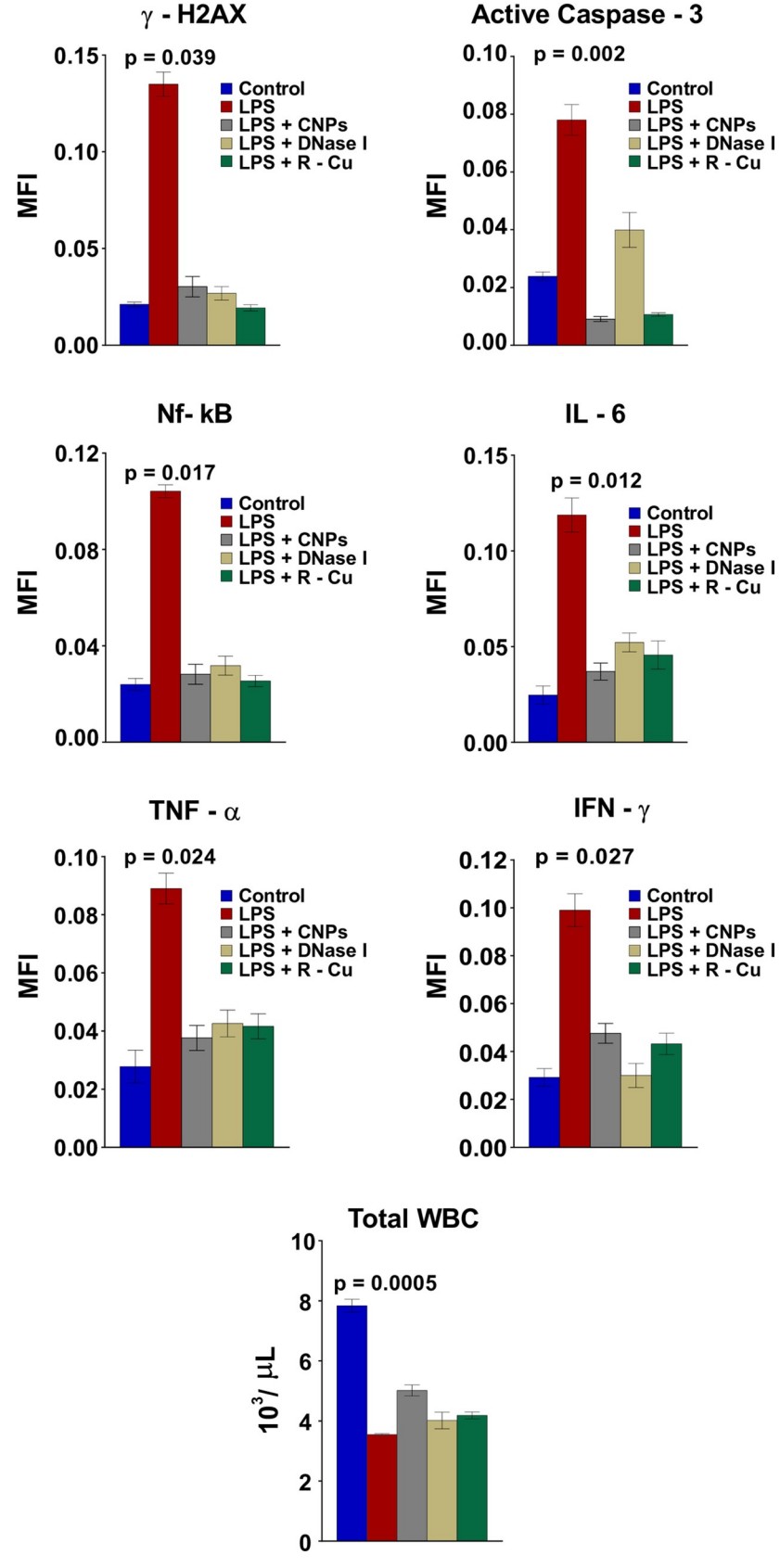

**Fig 4. DNA damage, apoptosis and inflammation in PBMCs following LPS treatment and their prevention by cfCh inactivating agents.** Treatment with LPS resulted in marked increase in DNA damage, apoptosis and inflammation in PBMCs which were significantly reduced by concurrent treatment with CNPs, DNase 1 and R-Cu. Reduction in WBCs count was also significantly ameliorated by concurrent treatment with above cfCh inactivating agents. DNA damage, apoptosis and inflammation were estimated by indirect immuno-fluorescence performed at 72 h post LPS. Methodological details are given under Material and Methods section. Each group in all experiments comprised of 5 animals and the histograms provide mean (± SEM) values. Mean (± SEM) values of N = 5 between groups were compared using non parametric one-way ANOVA (Kurskal—Wallis test) with Dunn's multiple comparison method at the significance and confidence level of p = 0.05.

apoptosis and inflammation in cells of lung, liver, heart, brain, kidney and small intestine following LPS challenge (Fig 5 and S3A–S3F Fig).

This was evidenced by marked activation of H2AX, active Caspase 3, NFκB, IL-6, TNFα and INFγ. The effects of CNPs, DNase I and R-Cu on these parameters was dramatic, and levels of all parameters were reduced to near normal levels. (Fig 5 and S3A–S3F Fig)

## Liver and kidney dysfunction and elevation of serum lactate and their prevention

Parenchymal DNA damage, apoptosis and inflammation is accompanied by liver and kidney dysfunction which are prominent features of sepsis [33]. Liver dysfunction following LPS treatment resulted in elevation of aspartate aminotransferase (AST), alanine aminotransferase (ALT) and lactate dehydrogenase (LDH). Levels of these enzymes were significantly reduced by concurrent treatment with CNPs, DNase I and R-Cu (Fig 6, upper panel).

Kidney dysfunction following LPS evidenced by elevation of creatinine and blood urea nitrogen (BUN) levels were also significantly attenuated by concurrent treatment with the three cfCh inactivating agents (Fig 6, middle panel). The elevated levels of serum lactate in LPS treated animals, indicative of tissue hypo-perfusion resulting from micro-vascular damage and organ dysfunction [34], was also significantly reduced by the three cfCh inactivating agents (Fig 6, lowest panel).

## Coagulopathy, fibrinolysis and low platelet count and their prevention

LPS treatment led to marked dysregulation of parameters of blood coagulation akin to that seen in disseminated intravascular coagulation [35]. A pro-thrombotic state was evident at 18 h post LPS with elevation of fibrinogen levels in serum [36], and decrease in levels of anti-thrombin [37] and protein C [38] (Fig 7, upper panel).

Abnormal levels of all three parameters could be reversed, except in case of anti-thrombin, by concurrent treatment of CNPs, DNase I and R-Cu (Fig 7, upper panel). We also observed increase in fibrinogen deposition in liver following LPS treatment apparently as a result of micro-vascular coagulopathy [39], which was significantly reduced by treatment with cfCh inactivating agents (Fig 7, middle panel and S4 Fig). Consumption of clotting factors apparently led to a fibrinolytic state with decrease in levels of thrombin-anti-thrombin complex (TAT complex)—a surrogate marker of plasma thrombin [40]—which was restored by the administration of cfCh inactivating agents (Fig 7, lowest panel, left hand image). Also evident was a drastic reduction in platelets count in LPS treated mice [41] suggesting a decrease in clotting capacity (Fig 7, lowest panel, right hand image). The low platelets count was significantly restored by treatment with CNPs and R-Cu, but not by DNase I.

## Fatality and its prevention

Finally, we show that treatment with cfCh inactivating agents can reduce fatality from LPS treatment (Fig 8).

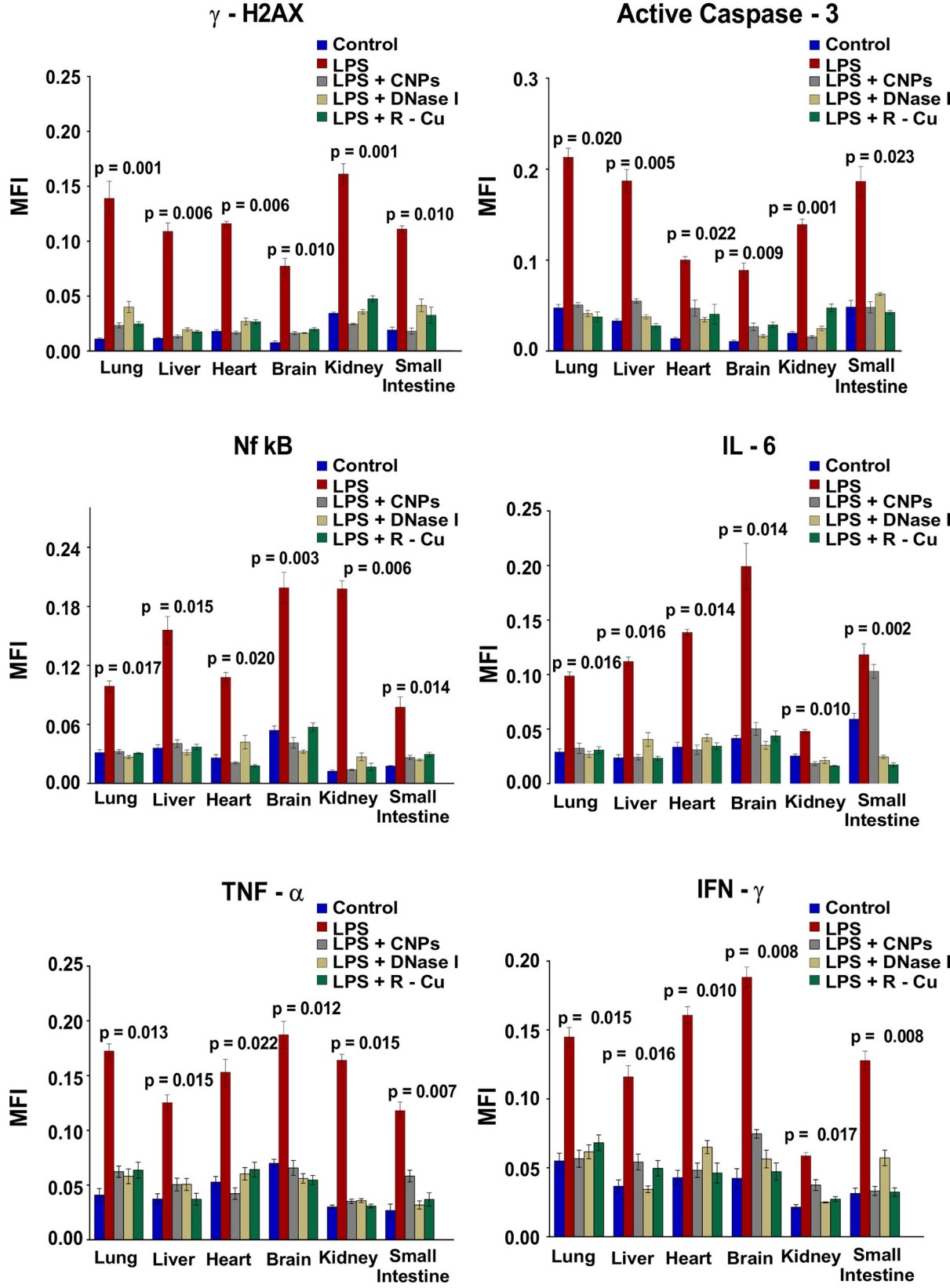

**Fig 5. DNA damage, apoptosis and inflammation in vital organs following LPS treatment and their prevention by cfCh inactivating agents.** Treatment with LPS resulted in marked increase in DNA damage, apoptosis and inflammation in multiple organs which were dramatically reduced by concurrent treatment with CNPs, DNase 1 and R-Cu. The above parameters were estimated by indirect immuno-fluorescence performed at 72 h post LPS. Methodological details are given under Material and Methods section. Each group in all experiments comprised of 5 animals and the histograms provide mean (± SEM) values. Mean (± SEM) values of N = 5 between groups were compared using non parametric one-way ANOVA (Kurskal—Wallis test) with Dunn's multiple comparison method at the significance and confidence level of p = 0.05.

Ninety percent of mice given a lethal dose of LPS died by day 10, while treatment with CNPs and R-Cu reduced the fatality rate to 50% whereas DNase I reduced it to 40%.

## Discussion

Patho-physiology of sepsis is complex and poorly understood [10]. Of date, no specific treatment for sepsis exists [10]. For long it was assumed that sepsis is caused by an intense inflammatory reaction to microbial infection [3]. However, failure of numerous clinical trials of anti-inflammatory agents to improve survival [10] has led to the conclusion that other unknown host factors might be responsible for the high mortality rates [18]. Recent research by our group suggests that cfCh from dying cells of the host might be one such factor [12–17]. cfCh released from dying cells, or those that circulate in blood, have the capacity to freely enter into healthy bystander cells to illegitimately integrate into their genomes and inflict dsDNA breaks [12–14,17]. The latter not only leads to apoptosis of cells [12, 13], but also to intense activation of inflammatory cytokines in the affected cells [15–17]. We have made the striking observation that fluorescent NFκB signals co-localize with those of γH2AX, which are activated at the sites of cfCh integration [12]. This has led to our proposal that inflammation is a direct consequence of dsDNA breaks induced by illegitimate integration of cfCh into the genome [15–17]. A close association of cfCh induced dsDNA breaks and inflammation was seen in multiple cell types *in vitro* [12] as well as in all organs and tissues examined *in vivo* [12]. cfCh inactivating agents not only prevented dsDNA breaks but also prevented inflammation [12]. In the present study, LPS injection led to copious release of cfCh particles from dying cells into extracellular spaces of vital organs (Fig 1A and 1B) and into the circulation (Fig 1C). We propose that a global activation of apoptosis and inflammation in all organs and tissues in response to LPS, including in cells of spleen, thymus and PBMCs, are a direct consequence of dsDNA breaks and inflammation induced by cfCh integration in their genomes leading to organ dysfunction and immune suppression. We hypothesise that trials of anti-inflammatory agents might have failed because they did not address the root cause of inflammation, i.e. genomic cfCh integration and dsDNA breaks, but rather were targeting the symptoms or consequences of it.

We undertook a separate experiment to demonstrate that cfCh inactivating agents themselves have no bystander damaging effects on host cells. Animals which were administered CNPs, DNAse I and R-Cu in the absence of LPS showed no DNA damaging effects (activation of H2AX) in brain cells of mice (S5 Fig).

Taken together, these findings support the theory that sepsis is caused by cfCh released from dying host cells following pathogen invasion leading to bystander host cell apoptosis and inflammation which are perpetuated in a vicious cycle with release of more cfCh from dying bystander cells. Clinical relevance of our finding is suggested by reports that blood cfCh levels are grossly elevated in patients with sepsis [42] and act as predictors of organ dysfunction and death [43]. It is pertinent to note here that we have proposed a similar vicious cycle to explain chemotherapy toxicity [20]. We have shown that toxicity of cancer chemotherapy is not so much due the damaging effects of the drugs themselves, but rather, like in case of sepsis, is due to release of cfCh from initial round of drug-induced cell death leading to a vicious cycle of

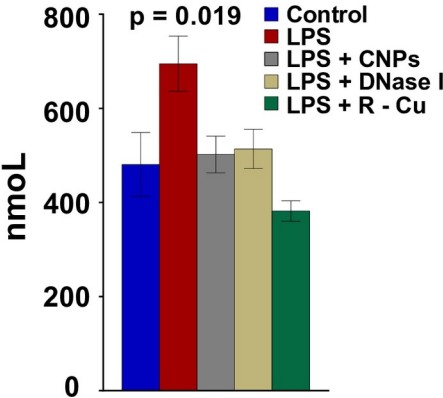

**Fig 6. Derangement of liver and kidney functions and elevation of serum lactate following LPS treatment and their prevention by cfCh inactivating agents.** Treatment with LPS resulted in marked increase in AST, ALT and LDH in the liver and creatinine and BUN in the kidney which were significantly reduced by concurrent treatment with CNPs, DNase 1 and R-Cu. Elevated serum lactate following LPS treatment was likewise significantly reduced following treatment with the cfCh inactivating agents. Liver and kidney functions were estimated by biochemical methods while serum lactate was estimated by colorimetric method, all performed at 72 h post LPS,. Methodological details are given under Material and Methods section. Each group in all experiments comprised of 5 animals and the histograms provide mean (± SEM) values. Mean (± SEM) values of N = 5 between groups were compared using non parametric one-way ANOVA (Kurskal—Wallis test) with Dunn's multiple comparison method at the significance and confidence level of p = 0.05.

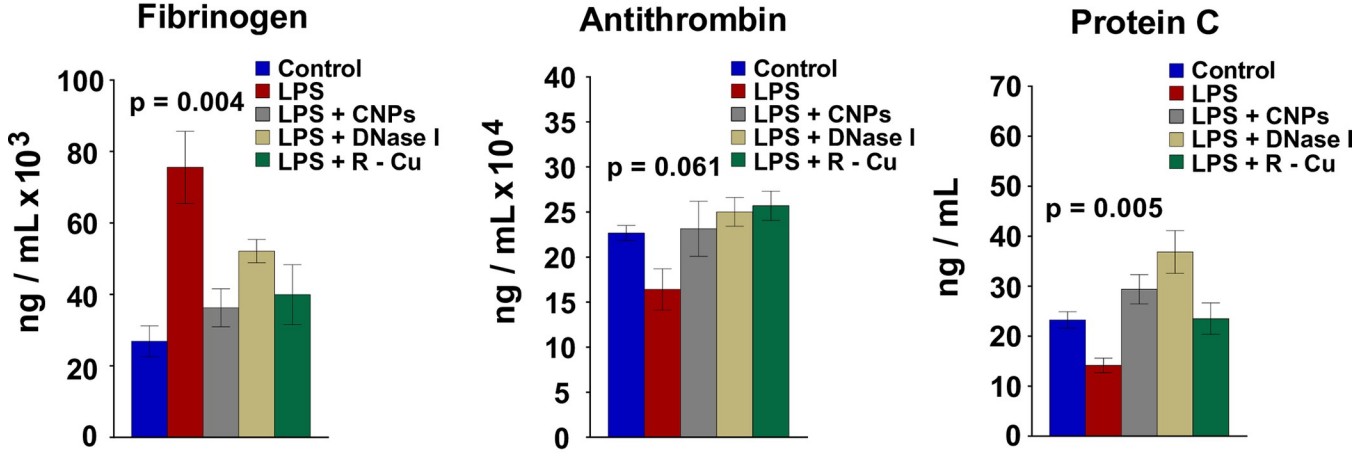

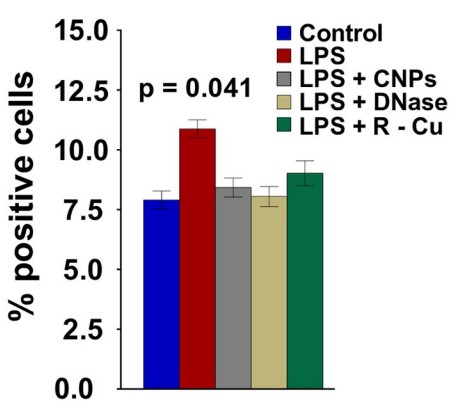

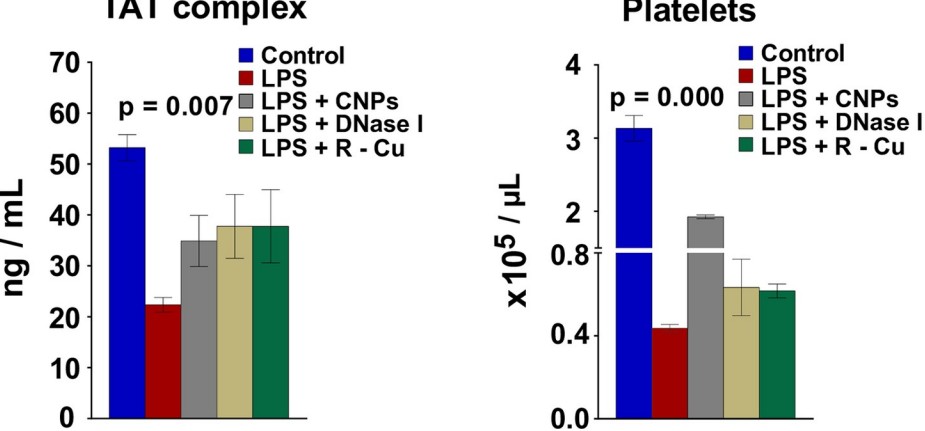

**Fig 7. Hyper-coagulation, fibrinogen deposition in liver, fibrinolysis and low platelets count following LPS treatment and their prevention by cfCh inactivating agents.** Treatment with LPS resulted in increase in serum fibrinogen levels and reduction in those of anti-thrombin and protein C. Concurrent treatment with CNPs, DNase 1 and R-Cu significantly reversed these changes in case of serum fibrinogen and Protein C, but not in case of anti-thrombin. Fibrinogen deposition in liver increased following LPS treatment which was significantly reduced by cfCh inactivating agents. The reduction in TAT complex and platelet count following LPS treatment was significantly reversed by treatment with CNPs, DNase 1 and R-Cu. Serum fibrinogen, anti-thrombin and

protein C and TAT complex were estimated by ELISA while fibrinogen deposition in the liver was measured by indirect immuno-fluorescence. Platelet count was performed by slandered procedure. All estimations were performed at 72 h post LPS. Methodological details are given under Material and Methods section. Each group in all experiments comprised of 5 animals and the histograms provide mean (± SEM) values. Mean (± SEM) values of N = 5 between groups were compared using non parametric one-way ANOVA (Kurskal—Wallis test) with Dunn's multiple comparison method at the significance and confidence level of p = 0.05.

DNA damage, apoptosis and inflammation perpetuated by more cfCh being released from dying cells of the host in a cascading manner [20]. We have also reported that bystander DNA damage and inflammation in healthy cells following radiotherapy is due to cfCh released from irradiated dying cells and that the bystander effect can occur both locally and in distant organs [25].

It has been reported that highly toxic extracellular histones are released in circulations in response to hyper-inflammatory challenge and can lead to endothelial dysfunction, organ failure and death from sepsis [44, 45]. The damaging effects of histones are mediated through Toll-like receptors 2 and 4 [46], and can be reduced by antibody to histones or activated protein C [44]. Neutrophil extracellular traps (NETs) are known to release cell-free DNA, histones as well as chromatin which have been implicated in sepsis [47, 48]. In this context, it should be

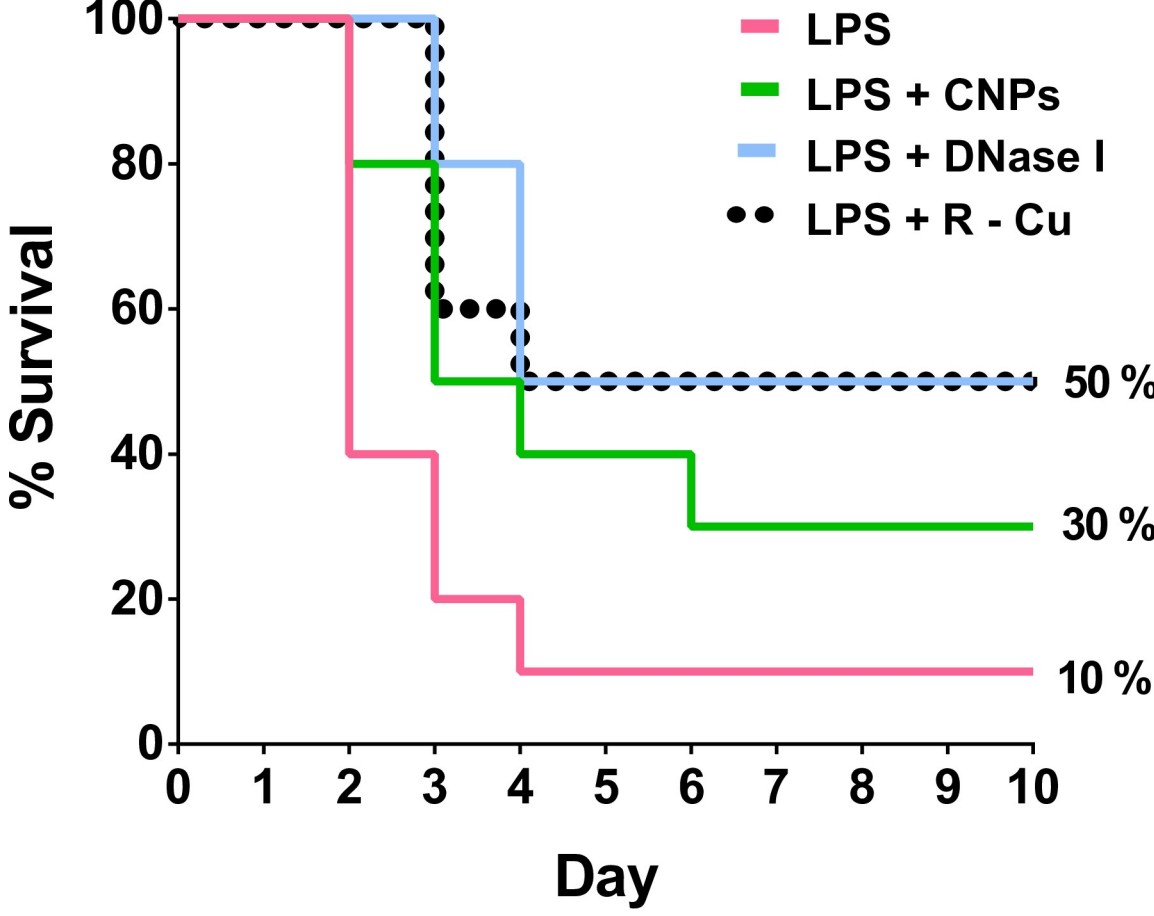

**Fig 8. Kaplan Meier survival analysis of LPS induced lethality and its prevention by cfCh inactivating agents (10 mice in each group).** Only one of 10 animals survived following i.p. injection of LPS while three animals survived following treatment with CNPs and 5 animals each survived after treatment with DNase 1 and R-Cu. The survival curves were compared using log-rank test with the use of PRISM Version 6.0.

noted from our results that inactivation of cfCh was more effective in preventing tissue damage and inflammation (Figs 3–5) and organ dysfunction (Fig 6), but was relatively less effective in reducing inflammatory cytokines in circulation (Fig 2) and in preventing fatality (Fig 8). Thus, it is possible that alarmin effects of histone and DNA and other nuclear proteins also contribute to endotoxin sepsis. However, the fact that sepsis inducing effects of cfCh were nullified to an equal extent by both anti-histone antibody complexed nanoparticles as well as anti-DNA agents such as DNase I and R-Cu suggest that these agents were inactivating a common target, viz cfCh.

The three cfCh inactivating agents not only had a global ameliorating effect on various parameters of LPS induced sepsis, but also improved survival rates of mice following a lethal dose of LPS (Fig 8). R-Cu and DNase I treatment achieved a survival figure of 50%, while in case of CNPs it was 30%, compared to 10% in LPS controls. CNPs, DNase I and R-Cu on their own had little bystander toxic side effects on healthy cells suggesting the possibility of their use in therapy of sepsis. It should be noted, however, that treatment with cfCh inactivating agents, in our experiments was started 4 h prior to LPS injection, unlike in patients with sepsis, wherein treatment is commenced only after signs and symptoms of sepsis have set in. Our study also does not exclude the possibility that host factors other than cfCh may be involved in LPS induced sepsis which remain to be identified [49]. Nonetheless, our results provide evidence for an association between cfCh released from dying host cells and the aetio-pathology of sepsis, suggesting a possible novel approach to treatment of sepsis with the use of cfCh inactivating agents.

## Conclusion

cfCh that are released from dying host cells in response to bacterial endotoxin may represent a global instigator of sepsis. cfCh inactivation may provide a novel approach to management of sepsis in humans.

## Supporting information

**S1 Fig. Representative immuno-fluorescence images of spleen and thymus showing activation of DNA damage, apoptosis and inflammation in splenocytes and thymocytes following LPS challenge and their prevention by cfCh inactivating agents.** The analyses were performed at 72 h post LPS. Methodological details are given under Material and Methods section.
(TIF)

**S2 Fig. Representative immuno-fluorescence images of PBMCs showing DNA damage, apoptosis and inflammation following LPS treatment and their prevention by cfCh inactivating agents.** The above parameters were estimated by indirect immuno-fluorescence performed at 72 h post LPS. Methodological details are given under Material and Methods section.
(TIF)

**S3 Fig. Representative immuno-fluorescence images of vital organs showing activation of DNA damage, apoptosis and inflammation following LPS challenge and their prevention by cfCh inactivating agents.** The analyses were performed at 72 h post LPS. Methodological details are given under Material and Methods section.
(TIF)

**S4 Fig. Representative immuno-fluorescence images of liver showing fibrinogen deposition following LPS challenge and its prevention by cfCh inactivating agents.** The analyses

were performed at 72 h post LPS. Methodological details are given under Material and Methods section.
(TIF)

**S5 Fig. Histograms to demonstrate that the three cfCh inactivating agents were themselves not toxic to mice.** Animals were divided into four groups: 1) control (n = 10) and those receiving 2) CNPs (n = 5), 3) DNase I (n = 5) and 4) R-Cu (n = 5) in doses as described in material and methods section. Animals were sacrificed on day 7 and their brain tissues were removed and cryo-sections were prepared for estimation of γ - H2AX by immunofluorescence as described in materials and methods section. The results show that the three cfCh inactivating agents did not lead to any increase in DNA damage in terms of H2AX activation.
(TIF)

**S1 Table. Cage side parameters for assessment of body weight to evaluate side effects in experiments lasting 72 h (10 mg/kg LPS).** Results show no change in body weight during the 72h period. Changes in physical activity were not monitored in these experiments.
(DOCX)

**S2 Table. A: Cage side parameters for assessment of body weight to evaluate side effects in the lethality experiment (20 mg/kg LPS).** Results show loss of body weight in LPS alone group but not in groups receiving LPS plus DNase I, R-Cu, and CNPs. B: Cage side parameters for assessment of physical activity to evaluate side effects in experiments in the lethality experiments (20 mg/kg LPS). Results show no loss of physical activity in control group. In LPs alone group loss of activity was observed. In LPS plus DNase, CNPs and R-Cu groups, a variable degree of recovery was observed.
(DOCX)

**S3 Table.** Analytical kits used and their procurement sources (Upper Table). Antibodies used and their procurement sources (Lower table).
(DOCX)

## Author Contributions

**Conceptualization:** Indraneel Mittra.

**Data curation:** Kavita Pal.

**Formal analysis:** Kavita Pal, Gorantla V. Raghuram.

**Funding acquisition:** Indraneel Mittra.

**Investigation:** Kavita Pal, Namrata Pancholi, Pritishkumar Tidke, Sophiya Siddiqui, Bhagyeshri Rane, Jenevieve D'souza, Alfina Shaikh, Saili Parab, Sushma Shinde, Vishal Jadhav, Soniya Shende, Gorantla V. Raghuram.

**Methodology:** Indraneel Mittra, Kavita Pal, Gorantla V. Raghuram.

**Project administration:** Indraneel Mittra, Kavita Pal.

**Resources:** Indraneel Mittra.

**Software:** Kavita Pal.

**Supervision:** Indraneel Mittra.

**Validation:** Kavita Pal.

**Visualization:** Kavita Pal, Gorantla V. Raghuram.

**Writing – original draft:** Indraneel Mittra.

**Writing – review & editing:** Indraneel Mittra.

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
