## [Decision Letter · Decision Letter 0]

18 Nov 2019

PONE-D-19-28396

Cell-free chromatin particles released from dying host cells are global instigators of endotoxin sepsis in mice.

PLOS ONE

Dear Prof. Mittra,

Thank you for submitting your manuscript to PLOS ONE. After careful consideration, we feel that it has merit but does not fully meet PLOS ONE’s publication criteria as it currently stands. Therefore, we invite you to submit a revised version of the manuscript that addresses the points raised during the review process.

Your manuscript was reviewed by two experts and we received mixed responses from them. Please address all comments. 

We would appreciate receiving your revised manuscript by Jan 02 2020 11:59PM. To enhance the reproducibility of your results, we recommend that if applicable you deposit your laboratory protocols in protocols.io, where a protocol can be assigned its own identifier (DOI) such that it can be cited independently in the future. For instructions see: http://journals.plos.org/plosone/s/submission-guidelines#loc-laboratory-protocols

We look forward to receiving your revised manuscript.

Kind regards,

Partha Mukhopadhyay, Ph.D.

Academic Editor

PLOS ONE

Journal Requirements:

** Please provide an amended Funding Statement that declares *all* the funding or sources of support received during this specific study ** (whether external or internal to your organization) as detailed online in our guide for authors at http://journals.plos.org/plosone/s/submit-now.  

Please state what role the funders took in the study.  If any authors received a salary from any of your funders, please state which authors and which funder. If the funders had no role, please state: "The funders had no role in study design, data collection and analysis, decision to publish, or preparation of the manuscript."

Reviewers' comments:

Reviewer's Responses to Questions

**Comments to the Author**

1. Is the manuscript technically sound, and do the data support the conclusions?

Reviewer #1: Partly

Reviewer #2: Yes

2. Has the statistical analysis been performed appropriately and rigorously? 

Reviewer #1: Yes

Reviewer #2: No

3. Have the authors made all data underlying the findings in their manuscript fully available?

Reviewer #1: Yes

Reviewer #2: Yes

4. Is the manuscript presented in an intelligible fashion and written in standard English?

Reviewer #1: Yes

Reviewer #2: Yes

5. Review Comments to the Author

Reviewer #1: In the present study “Cell-free chromatin particles released from dying host cells are global instigators of endotoxin sepsis in mice” the authors studied the possible role of cell-free chromatin in sepsis. This is a consecutive study after the discussion of cell-free chromatin particles in other disease models from the same authors. Here are some major concerns:

1. The authors claimed Figure 1A is the proof of the existence of cell free chromatin particles in LPS induced sepsis model. However, there are some major defects in these images. First, the images cannot reflect the typical structures of brain, lung and heart tissues. Second, the DNA antibody listed by the authors is mouse IgM, but the secondary antibody is against mouse IgG. Theoretically this immunostaining cannot work. Third, a lot of areas have positive DAPI staining but not DNA staining. Forth, the “positive” staining of cell free chromatin particles cannot be simply described as chromatin in the extracellular area. According to the data, they can only be described as somewhere not in the nucleus. Fifth, the specificity of these staining, especially the possibility that they are the result of unspecific binding of antibodies to dead tissues, should be better defined. The authors need to revise all these defects.

2. Although the authors used three reagents to neutralize or degrade cell free chromatin, necessary controls were not used. For neutralizing antibody, an isotype control Ig antibody should be used. For an enzyme, a deactivated enzyme should be used. For a compound containing heavy metal, the same organic compound with a light metal should be used.

3. In Figure 6, LDH is not a tissue specific parameter so the authors should not use it to define liver damage. Creatinine and BUN are kidney parameters but they are not produced by kidneys. So the authors should not put “renal” before them. Also, AST is not liver specific, so the authors should remove “liver” before AST.

4. The alarmin effects of DNA, histone and other nuclear proteins have been well defined. Do the authors claim a unique function of the cell-free chromatin particles as a whole or a combined phenomenon of the already defined alarmins?

Reviewer #2: In the study 'Cell-free chromatin particles released from dying host cells are global instigators of endotoxin sepsis in mice', the authors have shown convincing proof that chCh particles are involved in sepsis, and neutralizing them helps alleviate the adverse pathophysiological effects associated with it. This is an important study, and hopefully can help develop effective strategies to deal with sepsis. Some minor revisions are recommended below as follows:

1) Abstract (Line 29) and Introduction (Line 69): Please check sentence construction for '..those that circulated blood..'.

2) Introduction: It would be nice if the authors can elaborate more on the mechanisms by which cfCh particles induces DNA damage, apoptosis and activation of the inflammatory cytokine cascades. How are cfCh particles even transported from the extracellular domain into the cell? Elaborate more on findings from previous work (Refs 12-17).

3) Materials and Methods: Animals: Please include data about any side effects experienced by the animals in the study plus their weight in the manuscript. This is important in order to rule out any harmful effects on 'bystander' healthy cells which the authors mention, but don't discuss in detail later in the discussion section.

4) Materials and Methods: Detection of cellular DNA damage, apoptosis and inflammation in organs, tissues and PBMCs by IF: Please elaborate on the indirect immuno-fluorescence method used to estimate levels of gH2AX, caspase-3, Bcl-2, NF-kB, IL-6, TNFa and IFNg.

5) Materials and Methods: Statistical analysis: As multiple groups are being compared simultaneously, a simple t-test won't suffice. Please use ANOVA or a non parametric test such as Mann Whitney.

6) Results: Please eliminate all figure captions. This is unecessary, or can be used as Figure legend captions.

7) Results: Result titles/caption should reflect actual findings and not just what was done.

8)Results: Figure 3: How were thymocytes and splenocytes isolated?

9) Figures: Image quality is poor. Please upload high quality tiff images.

10) Discussion: Please discuss the potential of side effects by using all of these cfCh particle-inactivating agents on healthy cells. Elaborate on bystander effect.

6. PLOS authors have the option to publish the peer review history of their article (what does this mean?). If published, this will include your full peer review and any attached files.

Reviewer #1: No

Reviewer #2: No

---

## [Author Response · Author response to Decision Letter 0]

6 Jan 2020

Reviewers’ Comments to the Author

Reviewer #1:

1. First, the images cannot reflect the typical structures of brain, lung and heart tissues.

Our Response : We agree. For typical structures of brain, lung and heart to be reflected we need H&E stained slides. However, in that case immunohistochemical analysis will not be possible.

Second, the DNA antibody listed by the authors is mouse IgM, but the secondary antibody is against mouse IgG. Theoretically this immunostaining cannot work.

Our Response : We were aware of this. We ensured that the secondary antibody also reacted with mouse IgM. We quote from the vendor’s-Datasheet: “the secondary antibody reacts with Mouse IgG gamma chain, as well as the light chains from all Mouse immunoglobulin classes [including mouse IgM (sic)]” (Sigma-Aldrich, AP160R). 

This has now been clarified in the text (lines 202-206, page 9).

Third, a lot of areas have positive DAPI staining but not DNA staining.

Our Response : It needs to be pointed out that these are paraffin sections wherein the cut surfaces are not perfectly even. The unevenness is clearly seen in the magnified confocal images. Unevenness of paraffin sections is likely to cause the antibodies not being able to uniformly access the DNA / histone epitopes in the nuclei. 

This issue has been discussed in the text (lines 307-311, page 13). 

Fourth, the “positive” staining of cell free chromatin particles cannot be simply described as chromatin in the extracellular area. According to the data, they can only be described as somewhere not in the nucleus. 

Our Response : We agree, and we have made the necessary amendments in the text (lines 311-313 and line 316, page 13). 

Fifth, the specificity of these staining, especially the possibility that they are the result of unspecific binding of antibodies to dead tissues, should be better defined. 

Our Response : This may be a remote possibility. We have made the necessary amendments in the text (lines 313-315, page 13). 

2. Although the authors used three reagents to neutralize or degrade cell free chromatin, necessary controls were not used. 

For neutralizing antibody, an isotype control Ig antibody should be used. 

Our Response : The use of an isotype control Ig antibody would have been important had we been characterizing the specificity of anti-histone antibody complexed nanoparticles (CNPs) for the first time. We are past that stage. We have already published in detail characteristics of CNPs (Ref. 19) and even obtained a US Patent (U.S. Patent No. 9,096,655, Issued: August 4, 2015). The high specificity of CNPs was fully addressed in the publication / patent application. We do not think we need to re-invent the wheel for this paper. We have also published several papers using CNPs to neutralize cfCh in mice, without Ig antibody controls, which did not evoke any queries from reviewers (Ref. 12, 13, 20, 25). We do not think it is necessary to use negative controls in a study which uses a well characterized and well established antibody complex. 

For an enzyme, a deactivated enzyme should be used. 

Our Response : As the name suggests, DNase I degrades DNA. We do not think that it is necessary to use a deactivated enzyme as a negative control for such a specific enzyme which is universally accepted. We have also published several articles using DNase I to inactivate cfCh without any deactivated enzymes as a negative control and without any objection from reviewers (Ref. 12, 13, 20, 25).

For a compound containing heavy metal, the same organic compound with a light metal should be used.

Our Response : Again, we are not addressing as to which is the best metal to use along with Resveratrol in this paper. This was done in the past when we had undertaken extensive research to come to the conclusion that Resveratrol and Copper is the best combination to degrade genomic DNA (Ref. 24), and subsequently to show that it can degrade the DNA component of cfCh to inactivate it in mice without any comments from reviewers (Ref. 12, 20, 25). During our initial extensive research we had tried out various combinations of plant poly-phenols and heavy and light metals to ultimately conclude that R-Cu was the most active combination. Light metals such as Zinc were found not to be active in combination with Resveratrol. 

Nonetheless, we have now added a line to clarify that light metals are not active in combination with Resveratrol (lines 180-181, page 8). 

3. In Figure 6, LDH is not a tissue specific parameter so the authors should not use it to define liver damage. Creatinine and BUN are kidney parameters but they are not produced by kidneys. So the authors should not put “renal” before them. Also, AST is not liver specific, so the authors should remove “liver” before AST.

Our Response : We completely agree. The necessary corrections have been made in figure no. 6.

4. The alarmin effects of DNA, histone and other nuclear proteins have been well defined. Do the authors claim a unique function of the cell-free chromatin particles as a whole or a combined phenomenon of the already defined alarmins?

Our Response : Yes, we believe that the role of cell-free chromatin (cfCh) in sepsis is unique and independent of any alarmin effects that DNA or histones might have. The sepsis inducing effects of cfCh are nullified to an equal extent by anti-histone antibody complexed nanoparticles as well as anti-DNA agents such as DNase I and R-Cu indicating that they are inactivating a common target, viz cfCh. We do not deny that free DNA and / or histones may also have independent alarmin effects; but in many of our experiments the three cfCh inactivating agents reduced the sepsis biomarkers to near control levels suggesting that cfCh is the key instigator of sepsis (Figures 3,5 and 6 for example). We have addressed this issue in the penultimate paragraph of the discussion section of the paper. 

References: The references above pertain to those in the manuscript. 

Reviewer #2: 

1. Abstract (Line 29) and Introduction (Line 69): Please check sentence construction for '…those that circulated blood..'.

Our Response : We thank the reviewer for pointing out these minor errors. These have now been rectified.

2. Introduction: It would be nice if the authors can elaborate more on the mechanisms by which cfCh particles induces DNA damage, apoptosis and activation of the inflammatory cytokine cascades. How are cfCh particles even transported from the extracellular domain into the cell? Elaborate more on findings from previous work (Refs 12-17).

Our Response : Thanks are due again. This issue has now been elaborated in the introduction section (lines 73-88, Pages 3 & 4).

3. Materials and Methods: Animals: Please include data about any side effects experienced by the animals in the study plus their weight in the manuscript. This is important in order to rule out any harmful effects on 'bystander' healthy cells which the authors mention, but don't discuss in detail later in the discussion section.

Our Response : Thanks. The issue of side effects in terms of physical activity and weight loss have now been addressed in the text (lines 131-138 page 6 and supplementary tables 1 and 2a & 2b. 

With respect to harmful side effects (bystander effects), we especially undertook an experiment to assess the independent toxicity, if any, of the three cfCh inactivating agents and observed that CNPs, DNase I, and R-Cu have little toxicity of their own as assessed by activation of H2AX (DNA damage) in brain cells. The results are given in supplementary figure 5 and discussed in the text (lines 442-445, pages 18 & 19) and lines 482-483, page 20.

4. Materials and Methods: Detection of cellular DNA damage, apoptosis and inflammation in organs, tissues and PBMCs by IF: Please elaborate on the indirect immuno-fluorescence method used to estimate levels of gH2AX, caspase-3, Bcl-2, NF-kB, IL-6, TNFa and IFNg.

Our Response : Details of indirect immuno-fluorescence method has now been given in the methods section (lines 243 – 254, pages 10 and 11).

5. Materials and Methods: Statistical analysis: As multiple groups are being compared simultaneously, a simple t-test won't suffice. Please use ANOVA or a non- parametric test such as Mann Whitney.

Our Response : The reviewer is absolutely right. We have reanalyzed the data using ANOVA in all our experiments. The ANOVA analytical details are given in lines 287–294, page 12.

6. Results: Please eliminate all figure captions. This is unnecessary, or can be used as Figure legend captions.

Our Response : Figure captions have now been removed.

7. Results: Result titles/caption should reflect actual findings and not just what was done.

Our Response : The actual findings have now been included in all figure legends. 

8. Results: Figure 3: How were thymocytes and splenocytes isolated?

Our Response : Perhaps we have created this confusion. Thymocytes and splenocytes were not isolated, rather, the various parameters were assayed on histological sections of thymus and spleen. This issue has been clarified in the text (lines 330, 335, 340 - 341, page 14). 

9. Figures: Image quality is poor. Please upload high quality tiff images.

Our Response : We have now improved the quality of the figures.

10. Discussion: Please discuss the potential of side effects by using all of these cfCh particle-inactivating agents on healthy cells. Elaborate on bystander effect.

Our Response : In response to the reviewer’s query, we especially undertook an experiment to assess the independent toxicity, if any, of the three cfCh inactivating agents and observed that CNPs, DNase I, and R-Cu have little toxicity of their own as assessed by activation of H2AX (DNA damage) in brain cells. The results are given in supplementary figure 5 and discussed in the text (lines 442-445, pages 18 & 19) and lines 482 – 483, page 20.

---

## [Decision Letter · Decision Letter 1]

23 Jan 2020

PONE-D-19-28396R1

Cell-free chromatin particles released from dying host cells are global instigators of endotoxin sepsis in mice.

PLOS ONE

Dear Prof. Mittra,

Thank you for submitting your manuscript to PLOS ONE. After careful consideration, we feel that it has merit but does not fully meet PLOS ONE’s publication criteria as it currently stands. Therefore, we invite you to submit a revised version of the manuscript that addresses the points raised during the review process.

Your manuscript was reviewed by two experts and one reviewer raised  minor comments. A quick editorial decision will be taken after satisfactory revision and the manuscript will not send to reviewers.

We would appreciate receiving your revised manuscript by Mar 08 2020 11:59PM. To enhance the reproducibility of your results, we recommend that if applicable you deposit your laboratory protocols in protocols.io, where a protocol can be assigned its own identifier (DOI) such that it can be cited independently in the future. For instructions see: http://journals.plos.org/plosone/s/submission-guidelines#loc-laboratory-protocols

We look forward to receiving your revised manuscript.

Kind regards,

Partha Mukhopadhyay, Ph.D.

Academic Editor

PLOS ONE

Reviewers' comments:

Reviewer's Responses to Questions

**Comments to the Author**

1. If the authors have adequately addressed your comments raised in a previous round of review and you feel that this manuscript is now acceptable for publication, you may indicate that here to bypass the “Comments to the Author” section, enter your conflict of interest statement in the “Confidential to Editor” section, and submit your "Accept" recommendation.

Reviewer #1: (No Response)

Reviewer #2: All comments have been addressed

2. Is the manuscript technically sound, and do the data support the conclusions?

Reviewer #1: Yes

Reviewer #2: Yes

3. Has the statistical analysis been performed appropriately and rigorously? 

Reviewer #1: Yes

Reviewer #2: Yes

4. Have the authors made all data underlying the findings in their manuscript fully available?

Reviewer #1: Yes

Reviewer #2: Yes

5. Is the manuscript presented in an intelligible fashion and written in standard English?

Reviewer #1: Yes

Reviewer #2: Yes

6. Review Comments to the Author

Reviewer #1: The current version of “Cell-free chromatin particles released from dying host cells are global instigators of endotoxin sepsis in mice.” made a lot of improvement compared to the last version. Here is a minor concern the authors still have to address:

The authors claimed that the effect of cell free chromatin is unique from already known alarmin effect of histone and DNA because their neutralization can bring inflammation to baseline level. However, this is not entirely true according to the result of circulating inflammatory factors (Figure 2) and sepsis survival (Figure 8). The authors need to provide a better discussion of the contribution from each inflammatory element. Also the authors need to provide some clues why tissue inflammation is resolved while circulating inflammatory factors are still high.

Reviewer #2: (No Response)

7. PLOS authors have the option to publish the peer review history of their article (what does this mean?). If published, this will include your full peer review and any attached files.

Reviewer #1: No

Reviewer #2: No

---

## [Author Response · Author response to Decision Letter 1]

27 Jan 2020

6. Review Comments to the Author

Reviewer #1: The current version of “Cell-free chromatin particles released from dying host cells are global instigators of endotoxin sepsis in mice.” made a lot of improvement compared to the last version. Here is a minor concern the authors still have to address:

The authors claimed that the effect of cell free chromatin is unique from already known alarmin effect of histone and DNA because their neutralization can bring inflammation to baseline level. However, this is not entirely true according to the result of circulating inflammatory factors (Figure 2) and sepsis survival (Figure 8). The authors need to provide a better discussion of the contribution from each inflammatory element. Also the authors need to provide some clues why tissue inflammation is resolved while circulating inflammatory factors are still high.

Our response: The reviewer’s concerns have now been addressed on page 19, lines 451-460.

---

## [Editor Report · Decision Letter 2]

29 Jan 2020

Cell-free chromatin particles released from dying host cells are global instigators of endotoxin sepsis in mice.

PONE-D-19-28396R2

Dear Dr. Mittra,

We are pleased to inform you that your manuscript has been judged scientifically suitable for publication and will be formally accepted for publication once it complies with all outstanding technical requirements.

With kind regards,

Partha Mukhopadhyay, Ph.D.

Section Editor

PLOS ONE
---

## [Editor Report · Acceptance letter]

4 Feb 2020

PONE-D-19-28396R2 

Cell-free chromatin particles released from dying host cells are global instigators of endotoxin sepsis in mice. 

Dear Dr. Mittra:

I am pleased to inform you that your manuscript has been deemed suitable for publication in PLOS ONE. Congratulations! Your manuscript is now with our production department. 

With kind regards,

on behalf of

Dr. Partha Mukhopadhyay 

Section Editor

PLOS ONE